# Multi-Conditional Constraint Generative Adversarial Network-Based MR Imaging from CT Scan Data

**DOI:** 10.3390/s22114043

**Published:** 2022-05-26

**Authors:** Mingjie Liu, Wei Zou, Wentao Wang, Cheng-Bin Jin, Junsheng Chen, Changhao Piao

**Affiliations:** 1Automation School, Chongqing University of Posts and Telecommunications, Chongqing 400065, China; liumj@cqupt.edu.cn (M.L.); zouweizwzw@163.com (W.Z.); rickwang28574@163.com (W.W.); chenjunsheng@cqupt.edu.cn (J.C.); 2HUYA Incorporation, Guangzhou 511446, China; sbkim0407@gmail.com

**Keywords:** multi-conditional constraint generative adversarial network, medical image modal transformation, object re-identification, brain CT-MR image dataset

## Abstract

Magnetic resonance (MR) imaging is an important computer-aided diagnosis technique with rich pathological information. The factor of physical and physiological constraint seriously affects the applicability of that technique. Thus, computed tomography (CT)-based radiotherapy is more popular on account of its imaging rapidity and environmental simplicity. Therefore, it is of great theoretical and practical significance to design a method that can construct an MR image from the corresponding CT image. In this paper, we treat MR imaging as a machine vision problem and propose a multi-conditional constraint generative adversarial network (GAN) for MR imaging from CT scan data. Considering reversibility of GAN, both generator and reverse generator are designed for MR and CT imaging, respectively, which can constrain each other and improve consistency between features of CT and MR images. In addition, we innovatively treat the real and generated MR image discrimination as object re-identification; cosine error fusing with original GAN loss is designed to enhance verisimilitude and textural features of the MR image. The experimental results with the challenging public CT-MR image dataset show distinct performance improvement over other GANs utilized in medical imaging and demonstrate the effect of our method for medical image modal transformation.

## 1. Introduction

Magnetic resonance (MR) imaging [1] and computed tomography (CT) [2], both of which are suitable for the inspection of lesions in various tissues throughout the body, are commonly used as computer-aided medical imaging diagnostic techniques. Due to non-invasive, non-radiation, multi-contrast, and the fact that it contains more pathological information, MR imaging is more often referenced for the diagnosis of most diseases compared with CT imaging [3], especially on soft tissues, ligaments, and organs. However, since MR imaging should be finished in an airtight space with a strong magnetic field in about half an hour, the procedure for MR imaging is contraindicated for some patients with claustrophobia, cardiac pacemakers, and artificial joints [4]. By contrast, CT scanning does not need to be carried out in an airtight environment. It can also be finished within a few minutes, which is easier for patients to endure. Therefore, it is significant and valuable to develop a method that can estimate an MR image from its corresponding CT scan data. Both MR images and CT images, showing the anatomy and pathology of each tissue structure with a single channel, are digital images. However, they belong to different modalities due to different imaging principles [5]. Due to this, we can treat MR imaging from CT scan data as a medical image modal transformation problem, which is one of the fundamental topics in computer vision.

Image modal transformation refers to the generation of images from one domain to another domain under certain technical conditions [6,7]. The traditional computer-vision-based medical image modal transformation can be divided into two categories: learning-based methods and atlas-based methods. Learning-based methods construct non-linear mapping between the MR and CT image according to handcrafted feature extraction [8,9]. Atlas-based methods approximate a matrix between the MR image and atlas MR image by image registration [10,11,12], which can be used to warp the associated atlas CT image to estimate the query MR image. Since these two kinds of methods utilize handcrafted features to conduct medical image modal transformation, they are not general to different datasets.

Fortunately, with the development of large-scale visual datasets and increased computing power, convolutional neural networks (CNNs) [13], with their strong discriminative power and feature representation learning capabilities, have demonstrated record-breaking performance in computer vision tasks [14,15], including medical image modal transformation. Zhao et al. [16] modified U-Net [17] to synthesize an MR image from CT scan data. They trained the network using the paired CT-MR dataset (as shown in Figure 1a) by just minimizing the voxel-wise loss [18] between the synthesized image and the reference image, which results in blurry generated output. To solve this problem, Nie et al. [19] proposed a method that combines the voxel-wise loss with an adversarial loss in the generative adversarial network (GAN) [20], which is a new type of deep-learning-based generative model, to synthesize CT images from MR scan data. Combining voxel-wise loss with adversarial loss can improve the blurry synthesis problem. However, it highly depends on the availability of a large number of aligned CT and MR images, which is difficult to collect. In addition, compared with the paired CT-MR image data, most medical institutions have considerable unpaired CT-MR image data (as shown in Figure 1b) that are scanned for different purposes and radiotherapy treatments. Different from the methods [21,22,23] based on paired data, Kim et al. [24] proposed a learning method to discover cross-domain relationships using DiscoGAN, which does not require any explicit paired labels and can learn the relationships between datasets from different domains. Woltertink et al. [25] dealt with unpaired data with a CycleGAN model [26], which is an image-to-image translation model using unpaired data in the natural image field. Inspired by CycleGAN, Jin et al. [27] proposed MRGAN to use paired and unpaired data in a single model to overcome the context-misalignment problem. Jin et al. [28] focused on objective function design to construct a realistic and accurate synthetic MR image. The objective function they designed consists of adversarial, dual-cycle-consistent [29], voxel-wise, gradient difference [30], perceptual, and structural similarity terms to balance quantitative and qualitative losses. Due to dual-cycle-consistent structure, these two methods can be viewed as semi-supervised learning, both of which can apply paired and unpaired data to train the network. Li et al. [31] used L1 loss and L2 loss, based on U-Net, to generate MR images from CT images. However, the details and textures of the generated MR images were quite different from the real MR images. Therefore, it is an urgent problem to improve the fidelity of generated MR images. In addition, generating truer MR images by GAN is still a challenge due to the discrimination ability of the discriminative model. On one hand, compared with the generative model, the discriminative model is shallow, which results in imbalance in generation and discrimination in GANs. On the other hand, the discriminative model cannot further identify synthetic MR images when they are similar enough to the real one. Table 1 shows the comparison of GAN-based models for medical image modal transformation in the above research studies, including network structure and objective function. Compared with these methods, we optimized both the network structure and objective function to design a multi-condition constraint GAN, which can generate MR images with high quality. The major work of this paper can be summarized as follows: We treat MR image synthesis as an object re-identification problem and introduce cosine loss, which combines with voxel error and perception error as the model function.We modify the generator based on cycleGAN and design the discriminator based on PatchGAN under the constraint of the paired CT-MR dataset.

We design the function and modify the GAN architecture to optimize quality of the generated MR image. The remainder of this paper is organized into four sections. We introduce the working principle of GAN and conditional GAN in Section 2; Section 3 introduces our proposed method in detail, including multiple conditional constraint-based GAN structure and objective function design. Section 4 demonstrates the experimental results and contains a discussion on the specific comparison analysis; Section 5 gives the conclusion of this paper.

## 2. The Working Principle of GAN and cGAN

Standard GAN is introduced as a typical unsupervised learning method to train a generative model. As in Figure 2a, the framework of GAN contains a pair of competing models: a generative model *G* that captures the data distribution, and a discriminative model *D* that estimates the probability that a sample comes from the training data rather than *G*. To learn a generator distribution $p_{g}$ over data x, the generative model builds a mapping function from a prior noise distribution $p_{z}\;$to data space as $G\left(z\right)$. The discriminative model outputs a single scalar representing the probability that x comes from training data rather than$\;p_{g}$. *G* and *D* are trained simultaneously to adjust parameters for *G* and *D* to minimize 1−logD(G(z)) and logD(x), respectively, as if they are following the two-player min-max game with value function L(D,G):(1)$$\underset{G}{\min}\;\underset{D}{\max}L(D,G)=E_{x\sim P_{data}}[\log D(x)]+E_{z\sim P_{z}}[1-\log D(G(z))]$$

Standard GAN is an unsupervised learning model which cannot control the category of the generated image. However, it can be extended to a conditional model (cGAN) [32] if both the generative model and discriminative model are conditioned on some extra constrained information y as shown in Figure 2b.y, which is any kind of auxiliary information including data from other modalities, can be fed into both the generative model and discriminative model as an additional input layer to perform the conditioning. In the generative model, the prior noise $p_{z}$ and x are combined in joint hidden representation, and the adversarial training framework allows for considerable flexibility in how this hidden representation is composed. In the discriminative model, x and y are presented as inputs to a discriminative function. The objective function is also the two-player min-max game as in Equation (2): (2)$$\underset{G}{\min}\;\underset{D}{\max}L(D,G)=E_{x\sim P_{data}}[\log D(x|y)]+E_{z\sim P_{z}}[1-\log D(G(z|y))]$$

In cGAN, the input of the generative model can also be an arbitrary image [33,34] besides $p_{z}\;$[32]. In this paper, we use the CT image as input and its corresponding real MR image as constraint to design a special cGAN structure, so as to realize the modal transformation from CT image to corresponding MR.

## 3. Multi-Condition Constraint GAN Model

The modal transformation process of the image is reversible, which means CT and MR images can be converted to each other based on different generators with the corresponding constraint. Inspired by CycleGAN, this paper constructs a multi-conditional constraint GAN model that includes a generator, an inverse generator, and two discriminators. The proposed model optimizes the training model with the minimum error between the real and the generated MR image, which can improve the fidelity and detailed characteristics of the generated MR image. Since both the real and generated MR image belong to the same modality, we innovatively treat the real and generated MR image discrimination as an object re-identification problem. Due to this, cosine error fusing with original GAN loss is designed to enhance verisimilitude and textural features of the MR image.

### 3.1. Model Architecture

This paper builds a multi-conditional constraint GAN model to realize MR image generation with CT images as input. As shown in Figure 3, the model takes real CT images $I_{CT}$ as input and real MR images $I_{MR}$ as constraint to generate MR images $G_{MR}(I_{CT})$ by generator $G_{MR}$. On this basis, the $G_{MR}(I_{CT})$ is used as input with $I_{CT}$ as constraint to generate CT images $G_{CT}(G_{MR}(I_{CT}))$ by inverse generator $G_{CT}$. The discriminators $D_{MR}$ and $D_{CT}$ should not only distinguish the authenticity of the input MR and the CT image, but also discriminate whether the image has a corresponding relationship with the input image.

Both the generator and the inverse generator applied in our method include an encoder and a decoder where the encoder is used to extract features from the image, and the decoder is used to generate an image with the same scale as the input. This process is implemented separately through convolution and deconvolution operations. The generator and inverse generator in our proposed multi-conditional constraint GAN model are both improved based on the image transformation network [35], which produces impressive results in real-time style transfer and single-image super-resolution. The network contains two stride-one convolutions at the beginning and the end of the architecture, two stride-two convolutions, nine residual blocks [36], and two fractional convolutions with 0.5 stride. The nine residual modules are intended to deepen the network and expand the receptive field, so as to obtain more semantic information and extract detailed features of medical images. Each residual block includes two convolutions using 256 filters of 3 × 3 size with reflection padding, which effectively avoids the boundary artifacts and ensures the sharpness of the generated images. Instance normalization [37] and a rectified linear unit (ReLU) are followed to each convolution except the final one. The hyperbolic tangent (Tanh) [38] activation function follows the final convolution to guarantee the output is within [−1, 1]. The specific structure is shown in Table 2. 

The design of the discriminator is inspired by PatchGAN, proposed by Isola [39], which aims to classify small overlapping image patches rather than images. Compared with other discriminators, this patch-level discriminator has fewer parameters and can emphasize detailed information in local areas. The discriminator takes *N* × *N* fragments as input instead of the entire image so that it can pay more attention to the high-frequency information of the image, which in turn encourages the model to generate more realistic images. In the GAN model, the process of the generator to generate images is much more complicated than the process of the discriminator to discriminate the image authenticity. To balance the performance of the generator and the discriminator in the model, the discriminator designed in this paper belongs to a shallow network containing five convolutional layers. The specific structure is shown in Figure 4. In the second to fourth convolutional layers, normalization and non-linear processing based on leaky ReLU [40] are performed after each convolutional layer. The discriminator not only needs to distinguish the authenticity of the input MR image, but also needs to determine whether the MR image and the input CT image have a corresponding relationship.

### 3.2. Loss Function

Both networks in GANs are trained simultaneously. The discriminators $D_{MR}$ and $D_{CT}$ are used to estimate the probability of a sample coming from real data. The goal of the generator $G_{MR}$ is to generate an MR image using the CT image as input under the constraint of the real MR image. Whereas the $G_{CT}$ can be used to generate the CT image as a reversed constraint to impel the $G_{MR}$ to generate a more realistic MR image. Since the goal is to generate an MR image which should be similar to the real MR image in structure and detailed texture features, the adversarial losses are applied to the generator $G_{MR}$ and corresponding discriminator $D_{MR}$. The objective can be expressed as follows:
(3)$$\begin{array}{ll} L_{GAN}(G_{MR},G_{CT},D_{MR},D_{CT})=E_{I_{CT},I_{MR}\sim p_{data}(I_{CT},I_{MR})}[\log(D_{MR}(I_{CT},I_{MR}))]\\ \;\;\;\;\;\;\;\;\;\;\;\;\;\;\;\;\;\;\;\;\;\;\;\;\;\;\;\;\;\;\;\;\;\;\;\;{+E}_{I_{CT}\sim p_{data}(I_{CT})}[1-\log(D_{MR}(I_{CT},G_{MR}(I_{CT})))]\\ \;\;\;\;\;\;\;\;\;\;\;\;\;\;\;\;\;\;\;\;\;\;\;\;\;\;\;\;\;\;\;\;\;\;\;\;+E_{I_{CT},I_{MR}\sim p_{data}(I_{CT},I_{MR})}[\log(D_{CT}(I_{CT},I_{MR}))]\\ \;\;\;\;\;\;\;\;\;\;\;\;\;\;\;\;\;\;\;\;\;\;\;\;\;\;\;\;\;\;\;\;\;\;\;\;+E_{I_{CT}\sim p_{data}(I_{CT})}[1-\log(D_{CT}(I_{CT},G_{CT}(G_{MR}(I_{CT}))))] & \end{array}$$ where $I_{CT}$ and $I_{MR}$ are the real CT and MR images; $D_{MR}(I_{CT},I_{MR})$ and $D_{CT}(I_{CT},I_{MR})$ represent the probability of the MR and CT images coming from real data. $D_{MR}(I_{CT},G_{MR}(I_{CT}))$ and $D_{CT}(I_{CT},G_{CT}(G_{MR}(I_{CT})))$ represent the probability of those coming from the generated one.

Typically, adversarial loss can generate visually satisfactory MR images. However, the MR image modal transformation not only generates an MR image, but also renders it with richer pathological information corresponding to the content and features of the CT images. Therefore, the purpose of medical image modal transformation is to make the generated image contain more detailed texture features on the premise of similarity to the real image structure. For the paired data {$I_{CT}$,$I_{MR}$}, the generator $G_{MR}$ is tasked with generating realistic MR images that are close to the reference $I_{MR}$ of the input $I_{CT}$. Although the inverse generator $G_{CT}$ is not required as a final product, adding the same constraint to the $G_{CT}$ enables a higher quality of generated MR image. Considering the above factors comprehensively, the constraints on the similarity of image structure and detailed texture features reflected by the voxel loss and perception loss between the generated and the real CT image and the generated and the real MR image are defined as:(4)$$\begin{array}{l} L_{voxel}(G_{MR}(I_{CT}),G_{CT}(G_{MR}(I_{CT})))=E_{I_{CT},I_{MR}\sim p_{data}(I_{CT},I_{MR})}\left[|\right|I_{MR}-G_{MR}(I_{CT}){\left|\right|}_{1}]\\ \;\;\;\;\;\;\;\;\;\;\;\;\;\;\;\;\;\;\;\;\;\;\;\;\;\;\;\;\;\;\;{+E}_{I_{CT},I_{MR}\sim p_{data}(I_{CT},I_{MR})}\left[|\right|I_{CT}-G_{CT}(G_{MR}(I_{CT})){\left|\right|}_{1}] \end{array}$$
(5)$$\begin{array}{l} L_{perc}(G_{MR}(I_{CT}),G_{CT}(G_{MR}(I_{CT})))=E_{I_{CT},I_{MR}\sim p_{data}(I_{CT},I_{MR})}[\frac{1}{K}\sum_{j=1}^{K}\frac{1}{H_{j}W_{j}C_{j}}||\varphi_{j}(I_{MR})-\varphi_{j}(G_{MR}(I_{CT}))|{|}_{1}]\\ \;\;\;\;\;\;\;\;\;\;\;\;\;\;\;\;\;\;\;\;\;\;\;\;\;\;\;\;\;\;\;\;\;\;\;\;{+E}_{I_{CT},I_{MR}\sim p_{data}(I_{CT},I_{MR})}[\frac{1}{K}\sum_{j=1}^{K}\frac{1}{H_{j}W_{j}C_{j}}||\varphi_{j}(I_{CT})-\varphi_{j}(G_{CT}(G_{MR}(I_{CT})))|{|}_{1}] \end{array}$$
where φ represents the VGG16 [41] model used to extract perception features, and $\varphi_{j}(G_{MR}(I_{CT}))$ and $\varphi_{j}(G_{CT}(G_{MR}(I_{CT})))$ represent the semantic features of the generated MR image and CT image, respectively. $\varphi_{j}(I_{CT})$ and $\varphi_{j}(I_{MR})$ represent the semantic features of the real CT image and MR image. $H_{j}$, $W_{j}$, $C_{j}$ represents the height, width, and depth of the feature map of the jth convolutional layer of VGG16. K is the number of convolutional network layers.

In our work, the goal of the generator is to generate MR images using CT images as input under the constraint of the real MR image, which means the generated MR should be similar to the real MR image in structure and detailed texture features. Inspired by this, the discrimination between the real and the generated MR image can be viewed as object re-identification. Cosine loss function can push all samples away from the decision boundary towards their parametrized class mean direction. This means it can not only classify the object, but can also converge the features based on different classifiers, which is beneficial for internal-category classification.

Object re-identification of intra-class can be regarded as a classification problem among different individuals in the same object category. However, the traditional object classification is mostly inter-class classification, which usually maps objects randomly to the boundary. Inter-class classification does not require intra-class compactness and inter-class separation, which means it cannot achieve better object re-identification of intra-class. The von Mises–Fisher (vMF) distribution does better in exploring the intrinsic relationship between data posterior loss and prior distribution, which is suitable for object re-identification of intra-class as the potential distribution structure of data. In order to effectively discriminate the generated MR and real MR images, we propose to use vMF distribution and learn mapping to map input samples to the feature space ℝ. In addition, we explore the relationship between data posterior loss and prior distribution based on Bayesian theory. The prior distribution of samples in the feature space ℝ follows the vMF distribution:(6)$$p(y=k|f(x))=\frac{\exp(\kappa \cdot {\hat{w}}_{k}^{T}f(x))}{\sum_{c=1}^{C}\exp(\kappa \cdot {\hat{w}}_{c}^{T}f(x))}$$
where κ is shared concentration parameter, ${\hat{w}}_{k}^{T}f(x)$ represents the objective evaluation of samples, c is the number of category, and ${\hat{w}}_{c}^{T}f(x)$ represents the objective evaluation of different categories.

The vMF distribution is an isotropic probability distribution on a one-dimensional sphere that peaks around direction ${\hat{w}}_{k}$ and decays as the cosine similarity decreases. The corresponding cosine loss function can be written as:(7)$$L_{cos}=-\sum_{i=1}^{N}\log p(y_{i}=k|f(x_{i}))=-\sum_{i=1}^{N}\log\frac{\exp(\kappa \cdot {\hat{w}}_{k}^{T}f(x_{i}))}{\sum_{c=1}^{C}\exp(\kappa \cdot {\hat{w}}_{c}^{T}f(x_{c}))}$$
where N is number of samples.

This paper mainly involves real MR images and generated MR images, as well as real CT images and generated CT images within the class recognition. Therefore, the cosine loss function includes the cosine error between the real MR image and the generated MR image and the cosine error between the real CT image and the generated CT image:(8)$$\begin{array}{l} L_{cos}(G_{MR}(I_{CT}),G_{CT}(G_{MR}(I_{CT})))=E_{I_{CT,}I_{MR}\sim p_{data}(I_{CT,}I_{MR})}(L_{\cos}(I_{MR},G_{MR}(I_{CT})))\\ \;\;\;\;\;\;\;\;\;\;\;\;\;\;\;\;\;\;\;\;\;\;\;\;\;\;\;\;\;\;\;\;{+E}_{I_{CT,}I_{MR}\sim p_{data}(I_{CT,}I_{MR})}(L_{\cos}(I_{CT},G_{CT}(G_{MR}(I_{CT}))) \end{array}$$

Combining the above optimizations, the objective function of the MR modal transformation based on the multi-conditional constraint GAN model with CT image as the input can be defined as:(9)$$\begin{array}{l} L(G_{MR},G_{CT},D_{MR},D_{CT})=L_{GAN}(G_{MR},G_{CT},D_{MR},D_{CT})\\ \;\;\;\;\;\;\;\;\;\;\;\;\;\;\;\;\;\;\;\;\;\;\;\;\;\;\;\;\;\;\;\;+\lambda_{voxel}\cdot L_{voxel}(G_{MR}(I_{CT}),G_{CT}(G_{MR}(I_{CT})))\\ \;\;\;\;\;\;\;\;\;\;\;\;\;\;\;\;\;\;\;\;\;\;\;\;\;\;\;\;\;\;\;\;+\lambda_{perc}\cdot L_{perc}(G_{MR}(I_{CT}),G_{CT}(G_{MR}(I_{CT})))\\ \;\;\;\;\;\;\;\;\;\;\;\;\;\;\;\;\;\;\;\;\;\;\;\;\;\;\;\;\;\;\;\;+\lambda_{cos}\cdot L_{cos}(G_{MR}(I_{CT}),G_{CT}(G_{MR}(I_{CT}))) \end{array}$$
where $\lambda_{voxel}$, $\lambda_{perc}$, and $\lambda_{cos}$ are all hyperparameters, which are used to balance various objective functions. We aim to solve the:(10)$$G_{MR}=\arg\underset{G_{MR},G_{CT}}{\min}\underset{D_{MR},D_{CT}}{\max}L(G_{MR},G_{CT},D_{MR},D_{CT})$$

In summary, the MR modal transformation algorithm based on the multi-conditional constraint GAN model with CT images as input can be described in Algorithm 1.
**Algorithm 1:** MR imaging modality transformation algorithm with CT as input.**Input:** CT-MR dataset with corresponding relationship, initial model weight**Output:** model weight *W ** after training1: *for i* = 1 to *n*//*n* indicates the number of training sessions2: Randomly read data from the corresponding CT-MR dataset $\left\{I_{CT}^{i},I_{MR}^{i}\}\sim p_{data}(I_{CT},I_{MR}\right)$3: *if i*%3==0:4: Update the parameters $D_{MR}$ and $D_{CT}$ of the discriminator in the model through the objective optimization function $L(G_{MR},G_{CT},D_{MR},D_{CT})$5: End *if*6: Update the parameters $G_{MR}$ of the generator and $G_{CT}$ of the inverse generator in the model through the objective optimization function $L(G_{MR},G_{CT},D_{MR},D_{CT})$7: End *for*8: Return generator $G_{MR}$

## 4. Results and Analysis

The brain CT-MR dataset published by SZSPH is used to evaluate our proposed method, which includes 367 pairs of CT-MR data with corresponding relationships. We randomly select 257 pairs as training data, and the remaining 110 pairs as test data. All samples in the dataset are resized to 256 × 256. The experimental environment is Intel(R) Core(TM)i7-9700 CPU @ 3.20 GHz processor, 16 GB RAM, NVIDIA TITAN Xp. The multi-conditional constraint GAN was trained with mini-batch stochastic gradient descent (SGD) with a mini-batch size of one. The number of iterations is set as 25,700 (100 epochs) during the model training; all the networks are trained at a learning rate of 0.0002 in the first 12,850 iterations and linearly decay to zero in the following 12,850 iterations. For all experiments, the following empirical values were used to train the model: $\lambda_{voxel}=100$, $\lambda_{perc}=1$, $\lambda_{cos}=1$. 

### 4.1. Performance Evaluation Index

To evaluate the effect of the generated MR image, we use the mean absolute error (*MAE*) and root mean squared error (*RMSE*) of the voxels between the generated and the real MR image:(11)$$MAE=\frac{1}{N}\sum_{i=0}^{N-1}||I_{MR}(i)-G_{MR}(I_{CT}(i))||$$
(12)$$RMSE=\sqrt{\frac{1}{N}\sum_{i=0}^{N-1}(I_{MR}(i)-G_{MR}(I_{CT}(i)){)}^{2}}$$
where *N* is the number of image voxel values. In addition, peak signal-to-noise ratio (*PSNR*) is also one of the important indicators to evaluate the imaging quality of medical images:(13)$$PSNR=10\mathrm{lg}(\frac{MAX^{2}}{MSE})$$
(14)$$MSE=\frac{1}{N}\sum_{i=0}^{N-1}(I_{MR}(i)-G_{MR}(I_{CT}(i)){)}^{2}$$
where mean square error (MSE) is the mean square error between the generated and the real MR image. MAX=10, which is the maximum value of the pixel in the image. *PSNR* measures the ratio between the maximum possible intensity value and *MSE* of the generated and real MR images. *MAE*, *MSE*, *RMSE*, and *PSNR* are all based on aligned images. Although all samples have been processed for image alignment, it is very difficult to obtain CT-MR data with zero alignment error. Therefore, the structural similarity (*SSIM*) and person correlation coefficient (*PPC*) between the generated and real MR image should also be calculated, which are defined as follows:(15)$$\begin{array}{l} SSIM(I_{MR},G_{MR}(I_{CT}))=\frac{\left(2\mu_{I_{MR}}\mu_{G_{MR}(I_{CT})}+V_{1})(2\mu_{I_{MR}G_{MR}(I_{CT})}+V_{2}\right)}{\left(\mu_{I_{MR}}^{2}+\mu_{G_{MR}(I_{CT})}^{2}+V_{1})(\mu_{I_{MR}}^{2}+\mu_{G_{MR}(I_{CT})}^{2}+V_{2}\right)}\\ \;\;\;\;\;\;\;\;s.t.\;\;\;SSIM(I_{MR},G_{MR}(I_{CT}))\;\leq 1,\;\;\\ \;\;\;\;\;\;\;\;\;\;\;\;\;\;SSIM(I_{MR},G_{MR}(I_{CT}))\;=1\;if\;and\;only\;if\;I_{MR}=G_{MR}(I_{CT}) \end{array}$$
(16)$$\begin{array}{l} PCC=\frac{1}{N}\sum_{i=0}^{N}\frac{\left(I_{MR}(i)-\mu_{I_{MR}(i)})(G_{MR}(I_{CT}(i))-\mu_{G_{MR}(I_{CT}(i))}\right)}{\sigma_{I_{MR}(i)}\sigma_{G_{MR}(I_{CT}(i))}}\\ \sigma_{I_{MR}}\sigma_{G_{MR}(I_{CT})} \end{array}$$
where $V_{1}$ and $V_{2}$ are constants to prevent the divisor from being zero. $\mu_{I_{MR}}$, $\mu_{G_{MR}\left(I_{CT}\right)}$, $\sigma_{I_{MR}}$, and $\sigma_{G_{MR}\left(I_{CT}\right)}$ are the average value and standard deviation of the generated and the real MR image, respectively. The smaller the values of *MAE* and *RMSE*, the larger the values of *PSNR*, *SSIM*, and *PCC*, and the higher the fidelity of the generated MR image.

### 4.2. Comparison of Results under Different Objective Optimization Functions

To verify the effectiveness of different items in the objective optimization function, we use the ablation method to evaluate them. Table 3 shows the comparison of four objective optimization function structures: (1) objective optimization using only the traditional GAN model function $L_{GAN}$ for model training; (2) on the basis of $L_{GAN}$, the voxel loss $L_{voxel}$ item between the generated and the real MR image is added for model training; (3) on the basis of (2), the loss term $L_{perc}$ between the generated and the real MR image is added for model training; (4) on the basis of (3), the cosine loss term $L_{cos}$ between the generated and the real MR image is added for model training. 

In Table 3, the values of *MAE*, *RMSE*, *PSNR*, *SSIM*, and *PCC* are the average measures over the test set. It can be found that the system performance has been optimized with different items joined. The integrated objective optimization function obtains the best performance. Therefore, the various loss functions proposed in this paper perform well. Perceptual error characterizes the semantic error between the generated and real MR image, which can cause the generated MR image to reflect more detailed texture features. The cosine loss treats the real and generated MR image discrimination as object re-identification, which can improve the authenticity of generated MR images.

Besides the quantitative analysis of the evaluation indicators above, we can obtain more image details by direct observation of the generated MR images. Figure 5 below shows the visualization results of the generated MR images under different objective functions. From the red rectangle in Figure 5, we can observe that there are differences in complex structure and detailed texture features between different images. Some generated MR images are incomplete in structure and have obvious structural elements missing from real MR images. Other generated MR images have low structural similarity, and many internal texture trends are not displayed. There are also some generated MR images that have insufficient advanced semantic extraction, in which detailed texture features are not obvious. In contrast, each loss item has a certain utility, and the objective function that integrates all the loss items can significantly improve the quality of the generated MR images.

### 4.3. Comparison of Results under Different GAN Models

To prove the significance of the multi-conditional constraint GAN model proposed in this paper, we use the same dataset to train multi-channel GAN [22], DiscoGAN [24], Deep MR-to-CT [25], and MR-GAN [27], which are all applied to generate medical images. The average values of *MAE*, *RMSE*, *PSNR*, *SSIM*, and *PCC* among different algorithms are compared. As shown in Table 4, it can be found that the multi-conditional constrained GAN model proposed in this paper performs the best. The imaging rate is 70.51 ms, which is fast enough for generating the MR image. The model takes the generated MR image as input and the CT image is generated by the inverse generator when the reversibility of the generated image is fully considered. This can effectively improve the fidelity of the generated MR image.

To intuitively reflect the performance indicators, we also describe the indicators in the form of box diagrams. As in Figure 6, the dot on the left of the box diagram represents the distribution of each generated MR image. The green triangle in the box represents the average value of all generated images in various evaluation indicators. The red dotted line connection can clearly compare each model under different evaluation indicators. Figure 7 indicates that the method proposed in this paper is superior to other models in performance. The circles next to the box plots represent a single image slice from the test dataset. The top and bottom box limits were calculated from Q25 and Q75. The green triangles and the horizontal lines denote the average and the median. The range of the box plot whiskers is given by [Q25−1.5×(Q75−Q25),Q75+1.5×(Q75−Q25)]. Any data point that falls outside of this range is typically considered an outlier and is indicated by a red cross. The averages displayed in Table 4 indicate that our proposed method outperformed the other methods for all measures, with the lowest *MAE* and *RMSE* and the highest *PSNR*, *SSIM*, and *PCC*, thus further verifying the utility of our architecture.

Moreover, the superiority of the algorithm proposed in this paper can also be verified through the human visual mechanism. Figure 7 shows the test results of different randomly selected GAN models from the test set. It can be seen that the generated MR image based on our method can better reflect the detailed texture information of the imaged object with less noise. This shows the multi-conditional constraint GAN model has stronger image construction capabilities, which can better reflect the continuity, smoothness, and semantics of imaging features in image generation. As for the main reason: on the one hand, the reversibility of the cGAN model is fully considered in this paper. The CT image generated by the inverse generator reversely constrains the feature correspondence between the generated MR image and the input CT image. On the other hand, the representation of image semantic information by deep CNN, as well as the characteristics of the modal attributes of the generated and real MR image, is fully considered. Based on this, we construct an objective optimization function including the voxel error, perception error, and cosine error between the generated and real MR image. It is more effective to drive the model to generate MR images with clear detailed texture features and a high peak signal-to-noise ratio.

## 5. Conclusions

In this work, MR imaging is viewed as a machine perception problem. A multi-conditional constraint GAN for MR imaging from CT scan data is proposed. Considering reversibility of GAN, both generator and inverse generator are designed for MR and CT imaging, respectively, which can constrain each other and improve consistency between features of CT and MR images. In addition, we innovatively treat the real and generated MR image discrimination as object re-identification; cosine error fusing with voxel loss and perception loss is designed to enhance the fidelity and detailed texture feature representation of the generated MR image. Quantitative and qualitative experiments conducted on the challenging public CT-MR imaging dataset show distinct performance improvement over other GANs utilized in medical imaging and demonstrate the effectiveness of our method for medical image modal transformation.

## Figures and Tables

**Figure 1 sensors-22-04043-f001:**
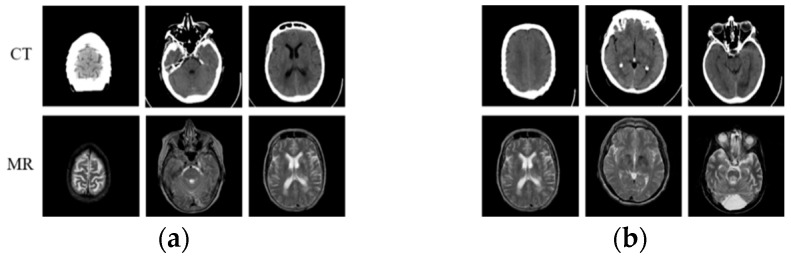
Paired and unpaired CT-MR dataset. (**a**) Paired CT-MR dataset; (**b**) unpaired CT-MR dataset.

**Figure 2 sensors-22-04043-f002:**
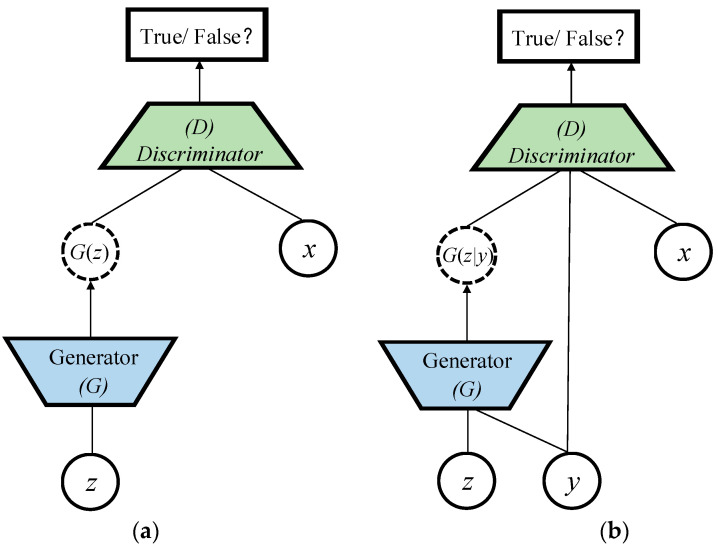
GAN and cGAN structure. (**a**) Standard GAN network model; (**b**) conditional GAN network model.

**Figure 3 sensors-22-04043-f003:**
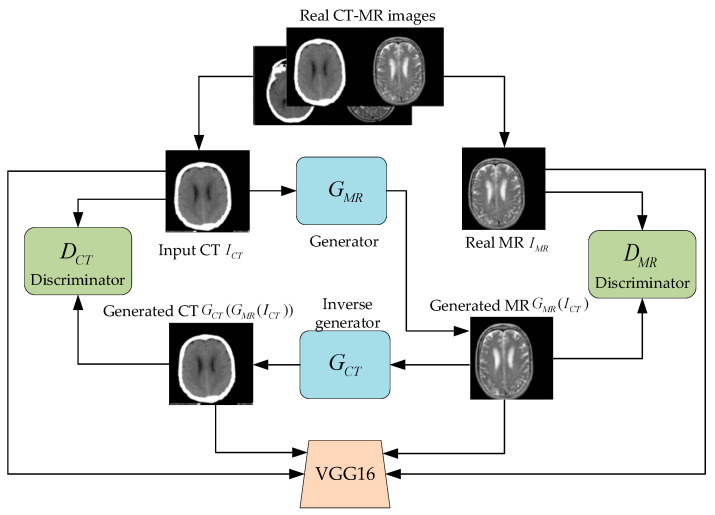
Multi-conditional constraint GAN structure for MR imaging using CT scan data.

**Figure 4 sensors-22-04043-f004:**
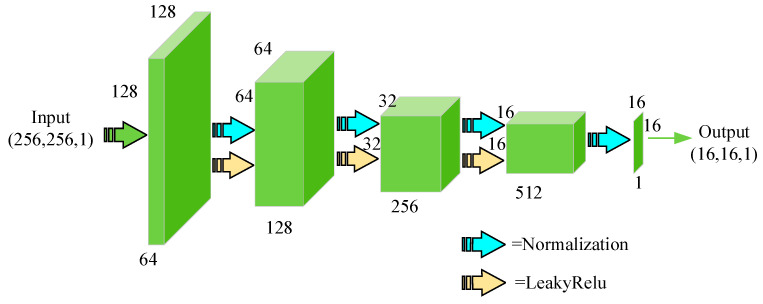
Flow diagram of the discriminator $D_{MR}$
and $D_{CT}$.

**Figure 5 sensors-22-04043-f005:**
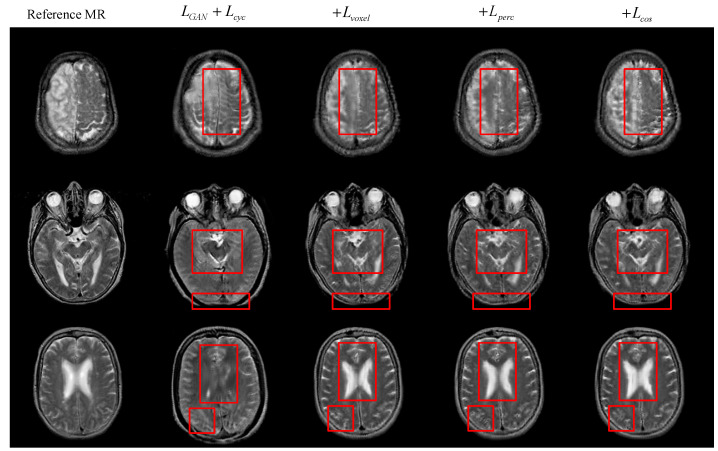
Qualitative comparison of the objective function.

**Figure 6 sensors-22-04043-f006:**
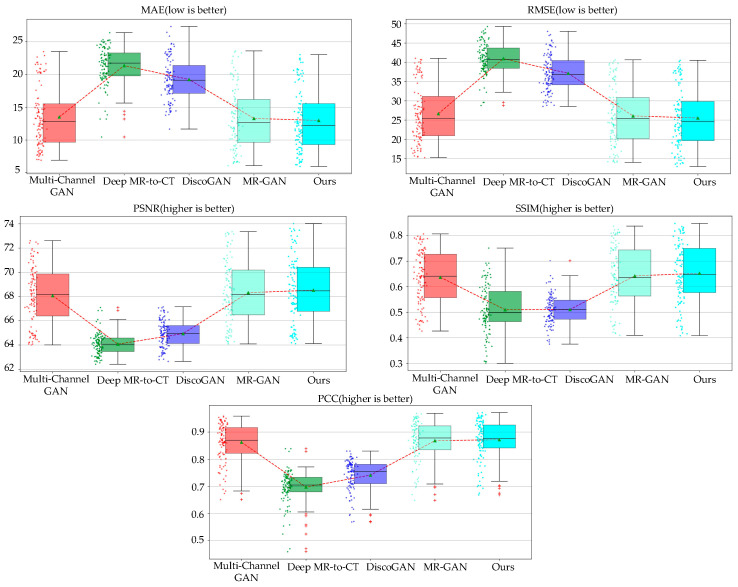
A comparison of the proposed method with baseline methods.

**Figure 7 sensors-22-04043-f007:**
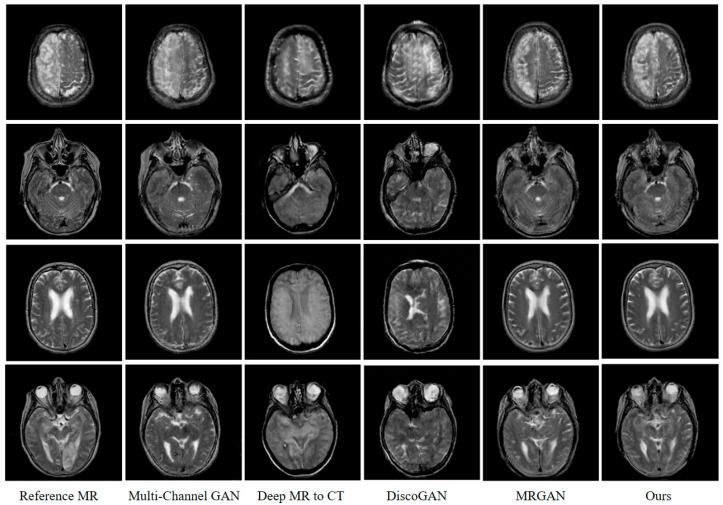
Qualitative comparison between our method and baseline methods.

**Table 1 sensors-22-04043-t001:** Comparison of GAN-based models for medical images modal transformation.

	Multi-Channel GAN [22]	Deep MR-to-CT [24]	DiscoGAN [25]	MR-GAN [27]
Model	Pix2Pix	cycleGAN	DiscoGAN	MR-GAN
Generator	U-Net	Residual Net	Customized	Residual Net
Number of Layers in Generator	16	24	8	24
Discriminator	Patch GAN	Patch GAN	Patch GAN	Patch GAN
Number of Layers in Discriminator	5	5	5	5
Objective function	Adversarial and voxel-wise	Least-squares adversarial and cycle-consistent	Adversarial and cycle-consistent	Adversarial, voxel-wise, and cycle-consistent

**Table 2 sensors-22-04043-t002:** Model architecture of generator $G_{MR}$ and inverse generator $G_{CT}$ layer name.

	Output Size	Filter Size/Stride	Number of Conv. Layers
Input image	H×W×1	\	\
Conv 1	H×W×64	7×7/1	1
Conv 2	H/2×W/2×128	3×3/2	1
Conv 3	H/4×W/4×256	3×3/2	1
Residual Block 1	H/4×W/4×256	3×3/1	2
Residual Block 2	H/4×W/4×256	3×3/1	2
Residual Block 3	H/4×W/4×256	3×3/1	2
Residual Block 4	H/4×W/4×256	3×3/1	2
Residual Block 5	H/4×W/4×256	3×3/1	2
Residual Block 6	H/4×W/4×256	3×3/1	2
Residual Block 7	H/4×W/4×256	3×3/1	2
Residual Block 8	H/4×W/4×256	3×3/1	2
Residual Block 9	H/4×W/4×256	3×3/1	2
Fractional Conv 1	H/2×W/2×128	3×3/0.5	1
Fractional Conv 2	H×W×64	3×3/0.5	1
Conv 4	H×W×1	7×7/1	1

**Table 3 sensors-22-04043-t003:** Ablation analysis of the objective function (↓ means that the smaller the value, the better, and ↑ means that the larger the value, the better). The best scores are displayed in bold.

	MAE ↓	RMSE ↓	PSNR ↑	SSIM ↑	PCC ↑
*L_GAN_*	19.054	36.343	65.157	0.52	0.761
*+L_voxel_*	13.163	25.844	68.423	0.647	0.869
*+L_perc_*	13.141	25.636	68.476	0.645	0.87
*+L_cos_*	**12.981**	**25.532**	**68.519**	**0.652**	**0.872**

**Table 4 sensors-22-04043-t004:** Results analysis of different GAN models. The best scores are displayed in bold.

	MAE ↓	RMSE ↓	PSNR ↑	SSIM ↑	PCC ↑	Imaging Rate
Multi-Channel GAN	23.513	26.647	68.076	0.637	0.862	25.74 ms
Deep MR-to-CT	21.362	40.941	64.063	0.51	0.697	51.47 ms
DiscoGAN	19.245	37.143	64.932	0.511	0.741	**22.81 ms**
MR-GAN	13.293	26.061	68.312	0.642	0.868	54.38 ms
Ours	**12.981**	**25.532**	**68.519**	**0.652**	**0.872**	70.51 ms

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
