# Peer review of "Multi-Conditional Constraint Generative Adversarial Network-Based MR Imaging from CT Scan Data"

_sensors, 2022, doi:10.3390/s22114043_

Round 1

Reviewer 1 Report

In this work, titled Multi-conditional generative adversarial network-based MR imaging from CT scan data, the authors propose and describe a multi-conditional generative adversarial network for MR imaging from CT scan data. The contribution of the paper consists of the fusion of cosine error with original GAN loss.

The paper has several linguistic errors, both from a grammatical and a syntactic point of view. I strongly encourage the authors to carefully proofread the paper, starting from the abstract, thus ensuring that the description of the work could be smoothly understood by the reader.

From a technical and scientific point of view, the paper needs several improvements to achieve the right level of clarity and completeness. In particular:

  • As for what concerns the Introduction, some information should be deepened. I suggest that authors make sure that all models and loss functions mentioned in the description of the state of the art are briefly explained in order to provide the reader with a complete picture. In addition, it would be better to place more emphasis on the novelty of the proposed work compared to previous ones and to provide a stronger explanation of its impact. In particular, the authors should explain in detail what re-identification of the object consists of.
  • The training process and setup are not described, which is a significant lack of reproducibility of the method. The authors should provide adequate information about several issues, such as:
    • which hyperparameters were chosen;
    • for how many epochs has the network been trained;
    • how was the average runtime for each result, or estimated energy cost;
    • how was the splitting of the training-/ validation- set done
  • In the Results section, it was not explicitly declared whether the reported quantitative scores were average or maximum measures over the test set.

Reviewer 2 Report

Comments

This study trained multi-conditional GAN model for generating MR images from CT imaging. The authors claimed the novelties where a cosine loss function is included to enhance verisimilitude and textual features of generated MR images. The experimental results look good. However, some explanations are not clearly explained and also there exist many typo and grammar errors in the manuscript. The following lists my comments in detail.

(1) The main concern is that why original MR images are also used as the input in order to generate MR images? To my understanding, if the real MR images are required in testing, it does not have contributions to train such kinds of GAN model.

(2) Abstract has many sentences that should be rephrased. For example, ‘MR imaging is an important computer aided diagnosis techniques’. Singular noun ‘technique’ should be used here. MR imaging is only one modality of medical imaging, which is not a computer aided diagnosis technique. Also it mentions that MR imaging has rich pathological information, which is also not convincing. Pathological information should be mostly observed in pathology slides.

(3) The sentence in lines 102-103 (‘Followed by summarizes and looks forward to this paper in section 5’) has the grammar error.

(4) In lines 141-142, CT and MR images can be converted to each other (not can convert to each other).

(5) Figure 3 is not clearly enough for illustration. It is mentioned that ‘As shown in Figure 3, the model takes real CT images ICT as input and real MR images IMR as constraints to generate MR images by generator GMR’. I did not see that the MR images are used as the constraints to the GMR. It is necessary to re-plot the figure 3 in order to make it become more clear for explanation.

(6) Figure 4 has a poor quality in resolution. Maybe it should be put into a table.

(7) Figure 5, why left and right sides are all input, without output?

(8) In equation (5), why VGG16 is applied on ICT, IMR? In figure (3), we could not see this information.

(9) There are so many different probability distribution functions. Why von-Mises-Fisher (vMF) distribution is used in equation (6)?

(10) In equation (9), it has several lambda values for weighting different loss functions. What are values used in experimental settings?

(11) in the line 298, it should be: which are defined as follows: Authors should check the manuscript carefully for so many grammar errors that are difficult to point one-by-one.

Round 2

Reviewer 2 Report

Comments

In the revised manuscript, the authors have tried to resolve the questions asked by the reviewers. The following lists two further comments to improve the manuscript quality.

(1) In the equation (9), λvoxel=100, λperc=1, λcos=1. It seems that the perception loss and cos loss have very small weights in the overall loss function. How these weighting numbers are empirically selected? As shown in Table 4, when perception and cos losses are added, the performances are slightly improved. How about using λperc=10, λcos=10, or other larger values?

(2) There are still some visible grammar errors in the manuscript. For example, in line 65 ‘to synthesis CT images from MR scan data’ should be ‘to synthesize CT images from MR scan data’. In line 67, ‘a large numbers of ’ should be ‘a large number of’. It is necessary to check the grammars carefully again for the whole manuscript.
